# Effects of minimum unit pricing for alcohol in South Africa across different drinker groups and wealth quintiles: a modelling study

Naomi Gibbs ,[1] Colin Angus ,[1] Simon Dixon,[1] Charles Parry,[2] Petra Meier[3]

[1]School of Health and Related Research, University of Sheffield, Sheffield, UK
[2]Alcohol, Tobacco and Other Drugs Research Unit, South African Medical Research Council, Cape Town, South Africa
[3]MRC/CSO Social and Public Health Sciences Unit, University of Glasgow, Glasgow, UK

**Correspondence to**
Ms Naomi Gibbs;
n.gibbs@sheffield.ac.uk

## ABSTRACT

**Objectives** To quantify the potential impact of minimum unit pricing (MUP) for alcohol on alcohol consumption, spending and health in South Africa. We provide these estimates disaggregated by different drinker groups and wealth quintiles.

**Design** We developed an epidemiological policy appraisal model to estimate the effects of MUP across sex, drinker groups (moderate, occasional binge, heavy) and wealth quintiles. Stakeholder interviews and workshops informed model development and ensured policy relevance.

**Setting** South African drinking population aged 15+.

**Participants** The population (aged 15+) of South Africa in 2018 stratified by drinking group and wealth quintiles, with a model time horizon of 20 years.

**Main outcome measures** Change in standard drinks (SDs) (12 g of ethanol) consumed, weekly spend on alcohol, annual number of cases and deaths for five alcohol-related health conditions (HIV, intentional injury, road injury, liver cirrhosis and breast cancer), reported by drinker groups and wealth quintile.

**Results** We estimate an MUP of R10 per SD would lead to an immediate reduction in consumption of 4.40% (−0.93 SD/week) and an increase in spend of 18.09%. The absolute reduction is greatest for heavy drinkers (−1.48 SD/week), followed by occasional binge drinkers (−0.41 SD/week) and moderate drinkers (−0.40 SD/week). Over 20 years, we estimate 20 585 fewer deaths and 9 00 332 cases averted across the five health-modelled harms. Poorer drinkers would see greater impacts from the policy (consumption: −7.75% in the poorest quintile, −3.19% in richest quintile). Among the heavy drinkers, 85% of the cases averted and 86% of the lives saved accrue to the bottom three wealth quintiles.

**Conclusions** We estimate that MUP would reduce alcohol consumption in South Africa, improving health outcomes while raising retail and tax revenue. Consumption and harm reductions would be greater in poorer groups.

## BACKGROUND

In South Africa (SA), there are high levels of reported abstinence coupled with high levels of binge drinking among those who do drink, resulting in significant levels of alcohol-related harm.[1] This harm is not distributed evenly throughout society with

### Strengths and limitations of this study

► This study presents the first epidemiological policy appraisal model of a minimum unit price applied to South Africa. Previous similar modelling work has been limited to high-income countries.

► Our model provides equity-relevant information by presenting results disaggregated by wealth quintiles, allowing for a more nuanced consideration of the potential impact of the policy.

► Our model has also benefited from a thorough programme of stakeholder engagement ensuring contextual and policy relevance.

► A key limitation is the alcohol pricing data, which is drawn from a relatively small sample in one locality. Further research would benefit from improved pricing data, specifically the different prices paid for alcohol by different population groups.

► Second, the model has not explored the financial impact on the poorest groups beyond increased alcohol expenditure. Further research should include exploration of broader financial benefits such as reduced private expenditure on healthcare or improved labour market outcomes.

the lower socioeconomic groups experiencing higher levels of harm, particularly for infectious diseases such as HIV.[2] The periodic prohibition of alcohol during the COVID-19 lockdown demonstrates political leaders' acceptance that alcohol causes harm to SA and signals a potential willingness to take strong action.[3] Provincial governments, such as the Western Cape, are considering a number of alcohol policy approaches, including the introduction of minimum unit pricing (MUP).[4]

MUP is a policy whereby a legal floor price is introduced, below which a fixed volume of ethanol cannot be sold to the public. It has already been introduced in several areas, which experience high levels of alcohol harm, including Scotland and Australia's Northern Territory. Evidence suggests that MUP has

been effective at reducing alcohol consumption, particularly among the heaviest drinkers, as they commonly drink the very cheap alcohol targeted by this policy.[5 6]

A limitation of transferring the current evidence for MUP is its focus on high-income countries. Transferring this evidence to SA would be problematic as it has very different drinking patterns, a very different harm profile with infectious disease and injury contributing significantly to the burden of alcohol, it has an informal sector, which is challenging to capture and it has very high levels of income inequality likely to result in differential baseline prices and price responsiveness.

The current alcohol landscape is rooted in the country's recent political history. In 1926, apartheid legislation prohibited African and Indian access to licensed premises or employment by licence holders. As a result, when the democratically elected government took power in 1994 they inherited a significant number of shebeens. Shebeens are (largely) unlicensed bars or pubs, found in townships, often open late and with a reputation for violence and risky sexual behaviour. Homebrew (mainly beer made from sorghum or other ingredients such as pineapple) can be purchased from shebeens along with other types of branded alcohol supplied by large alcohol manufacturers and mainly distributed through larger licensed outlets using bulk discounts. Although beer is the most popular drink, the consumption of large quantities of cheap wine is also prevalent and can be linked back to farm labourers being paid in cheap wine.[7]

The South African government currently use alcohol excise tax to compensate for some of the social costs they attribute to alcohol consumption.[8 9] The system is based on targets for the proportion of the price that constitutes tax (excise tax plus value-added tax (VAT)). This varies by drink type with wine lowest followed by beer then spirits. The government has indicated a willingness to innovate and pursue public health improvements via fiscal policy with the introduction of a sugar tax in 2018. However, in a country with high levels of socioeconomic inequality, there are concerns regarding possible financial impact of pricing policies on the poorest groups.[10] Evidence on public health pricing policies often fails to consider distributional impact by income-groups.[11]

When designing public health economic models for unique policy contexts, ongoing engagement with local stakeholders is essential. The purpose of engagement is twofold: to shape the direction of the research using expert local knowledge (including understanding the problem, guiding model development and ensuring policy relevance) and to provide channels for communication creating potential for the evidence to contribute to policy design.[12–14]

We aimed to: (1) present estimates of the change to alcohol consumption, individual expenditure, retail and tax revenue following the introduction of a South African MUP, using a purpose built model, (2) estimate the impact on a limited number of alcohol-related health conditions and associated healthcare costs, (3) explore the potential equity implications via the demonstration of impact by both drinker group and wealth quintile, (4) highlight parameters that are particularly influential to the results and areas that require further research.

## METHODS

We built an epidemiological policy appraisal model coded in R (code available here), using a comparative risk assessment approach with multistate life tables.[15] A stakeholder mapping exercise was carried out following scoping conversations with three academic experts from three South African institutions. Following this, a shortlist of policy professionals, civil society members and local academics was drawn up and checked via the scientific and ethical review process. They were engaged via scoping interviews and three workshops, at the beginning, middle and end of the modelling process. Stakeholders informed key decisions including the specific policy to simulate, levels of the MUP, health outcomes of interest, assumptions on homebrew switching behaviour and validation of our choice of data sources.

Two distinct sections of the model were defined (figure 1):

I. Price to consumption: baseline prices were estimated for drinker groups (heavy drinkers, occasional binge drinkers, moderate drinkers) and wealth groups. Consumption was estimated at the individual level, this includes the proportion of alcohol drunk that is homebrew. Following a change in price, the new price and subsequent consumption levels were estimated. This accounts for both mean and peak weekly alcohol consumption.

II. Consumption to harm: the relationship between mean and peak consumption and alcohol-related harm and associated costs were estimated.

There is no single data set that can provide all the required data for the model and, thus, a combination of survey data sets, market research data, and evidence from published literature were used (figure 2).

### Price to consumption
#### Baseline consumption and prices
Our model started by estimating mean and peak alcohol consumption at current alcohol prices at the individual level. We categorised drinkers into three exhaustive and mutually exclusive groups; moderate (less than 15 standard drinks (SDs) per week); occasional binge (less than 15 drinks per week but more than 5 on one occasion) and heavy (15 or more drinks per week). An SD in SA is currently 15 mL or 12 g of pure ethanol. We generated price distributions for wealth and drinker groups using real price data linked to individual drinking from the International Alcohol Control Study (IAC)[16] survey 2014/2015 completed in the metropolitan district of Tshwane. The IAC asked for highly detailed data about prices in both

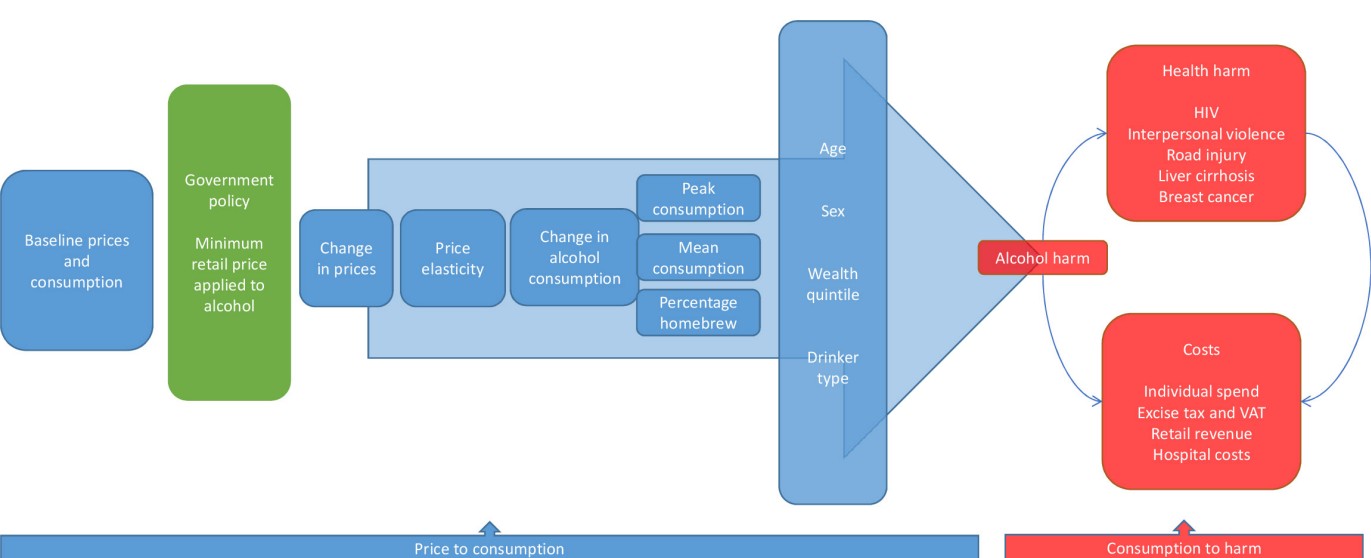

**Figure 1** Conceptual model framework.

on-trade and off-trade locations and took into account container size, drink type and number of drinks purchased. Alcohol was treated as one commodity as the 863 price observations were distributed between drinker and wealth groups instead of by alcohol type. Wealth quintiles were chosen as our measure of socio-economic status as income was not available in the pricing data set, whereas asset ownership and common demographic data such as age, sex and education were available. A detailed description of the above is given in the appendix parts 1 to 6.

## Applying an MUP

A government policy of legislating for an MUP of R5, R10 and R15 per South African SD was introduced. Prices below the MUP threshold were increased to the threshold, while products above were unaffected. We did not include prices for homebrew. The distribution of prices faced by each wealth/drinker group was used to calculate the mean price per SD before and after the policy. This then provided a percentage change in the mean price (table 1).

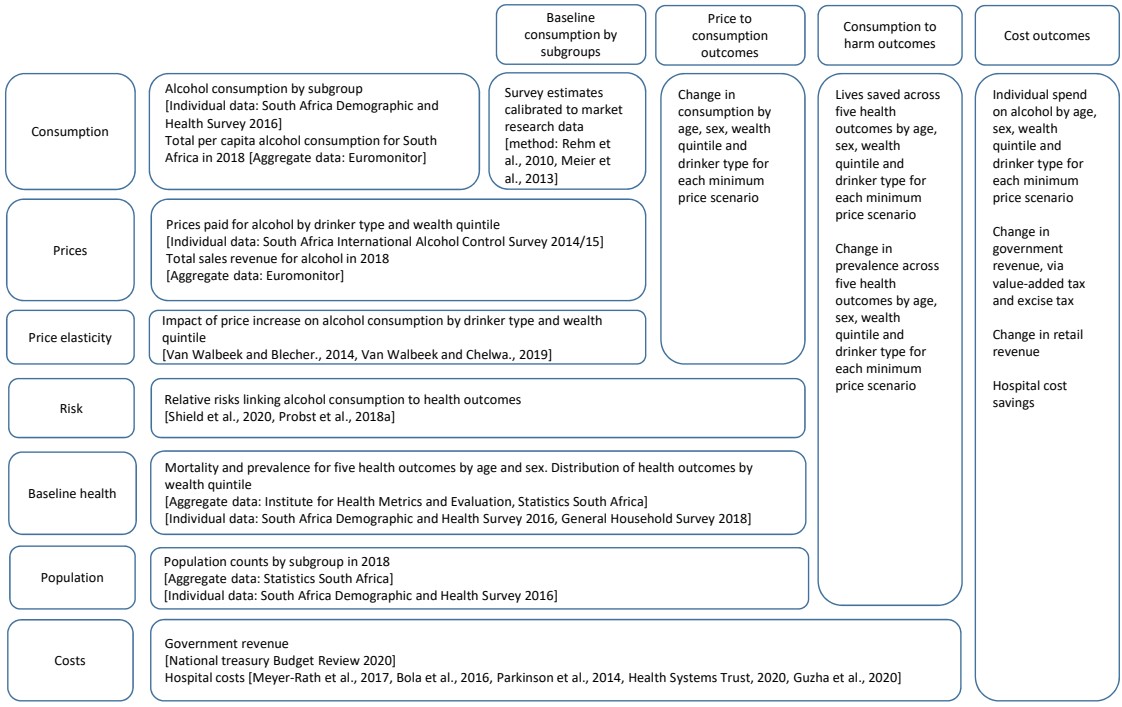

**Figure 2** Data inputs for model.

 

**Table 1** Price and elasticity data inputs by wealth quintile and drinker group

|  | Q1 | Q2 | Q3 | Q4 | Q5 |
|---|---|---|---|---|---|
| **Baseline price per standard drink** | | | | | |
| Moderate | R9.13 | R9.13 | R9.13 | R11.6 | R11.6 |
| Occasional binge | R7.97 | R10.0 | R10.1 | R13.4 | R11.1 |
| Heavy | R7.78 | R9.65 | R9.23 | R10.6 | R12.8 |
| **Percentage change in mean price following R10 MUP** | | | | | |
| Moderate | 22% | 22% | 22% | 20% | 20% |
| Occasional binge | 37% | 16% | 24% | 11% | 19% |
| Heavy | 33% | 26% | 25% | 24% | 21% |
| **Price elasticities used in the model** | | | | | |
| Moderate | −0.53 | −0.53 | −0.31 | −0.31 | −0.31 |
| Occasional binge | −0.29 | −0.29 | −0.17 | −0.17 | −0.17 |
| Heavy | −0.24 | −0.24 | −0.14 | −0.14 | −0.14 |

*Standard drink in South Africa defined as 15ml or 12 grams of pure ethanol
MUP, minimum unit pricing; Q1, poorest.

### Elasticity of demand for alcohol

The change in price was translated into a change in individual consumption using an elasticity of demand for alcohol. We used previously published elasticities for SA, calculated separately by drinker group: −0.4 to −0.22, −0.18 for moderate, occasional binge and heavy drinkers, respectively,[4] and adjusted for wealth quintile using additional evidence from SA[17] (table 1) (online supplemental appendix 7).

Those who drank both recorded alcohol and homebrew dampened the policy impact by switching some of their drinking to homebrew. Stakeholders indicated that 30% of the reduction in recorded alcohol could be assumed as being compensated for via an equivalent increase in homebrew. This was varied between 0% (no switching) and 100% (full switching) in the sensitivity analysis.

### Individual spend, tax and retail revenue

Total retail spend was computed by aggregating population-weighted individual spend. This figure was increased by 1.25 (100/80) as consumption was calibrated to 80% of official sales volume data.[18]

As an MUP is applied before VAT is calculated, we estimated VAT as 15% of the total aggregate spend. Excise tax was calculated by starting with the total 2018 alcohol excise tax revenue from the Treasury Budget Report.[9] This was adjusted by percentage change in volume of alcohol sold (we used a fixed ratio between volume and excise tax). Retail revenue was calculated by taking VAT and excise taxation away from total spend (online supplemental appendix 8).

### Consumption to harm

#### Relative risks and potential impact fractions

We used published estimates of relative risks associated with different levels of alcohol consumption (online supplemental appendix 9). We then used these to calculate relative risks for each outcome for each individual.

We used potential impact fractions (PIFs), a widely used approach in epidemiological modelling, to estimate the impact of a change in exposure to risk on a change in outcomes.[19] We incorporated population weights and computed the PIFs by sex, wealth group and drinker groups (online supplemental appendix 10).

#### Baseline health

Baseline deaths and cases (population prevalence) of the five disease and injury conditions (HIV, road injury, intentional injury, liver cirrhosis and breast cancer) were apportioned by drinker group, sex and wealth quintile. The probability of death for each disease was calculated for baseline and taken away from overall probability of death for each single year of age given in the life table to give a probability of death from non-modelled causes. This probability of death from non-modelled causes remained constant at every policy scenario. The probability of death from the five diseases of interest then varied according to the policy level and the corresponding PIF. A more detailed description is given in online supplemental appendices 11 and 12.

#### Projecting the population

We modelled counterfactual population structure (ie, in the absence of the policy) over 20 years, starting from 2018.[20] We created multistate life tables in which the population faces a probability of mortality for each of the five disease/injury conditions and for non-modelled causes each year. The model generated alternative population impact fractions (as above) for baseline and for each policy scenario. Using the relevant population impact fraction and rerunning the multistate life table enabled a calculation of the difference between baseline and the policy. HIV, road injuries and intentional injuries realise the full impact of the reduction in drinking immediately, whereas the health impact on liver cirrhosis and breast cancer are subjected to lags in the effect[21] (online supplemental appendix 13).

The life tables for the 20-year time horizon were used in combination with the probability of having the disease and the PIFs under each policy, to estimate the number of cases.

#### Hospital costs

Prevalence of disease/injury at each policy scenario for each year of the model was multiplied by the proportion who receive hospital treatment and the relevant hospital cost applied (online supplemental appendix 14). We converted all costs to 2018 prices using the Consumer Price Index.[22] Future costs were discounted at 5%.

## Sensitivity analysis

We explored key uncertainties in the model using scenario analysis informed by previous published alcohol modelling work,[23] our knowledge of the limitations of the data and stakeholder input. For each alternative scenario, relevant results were compared with central estimates. The key parameters explored were elasticities, proportion of abstainers, HIV baseline estimates, socioeconomic gradients of health, proportion of switching to homebrew and discount rates for costs (online supplemental appendix 15).

### Patient and public involvement

Patients were not involved in this study.

## RESULTS

### Estimated consumption and spend impact

Our findings are presented primarily for an R10 MUP, but with some comparisons across all three pricing levels. The policy appraisal results are reported by quintile further disaggregated by drinker group (table 2). In the model, drinking prevalence increases with wealth (27% up to 38%) as does the prevalence of heavy drinking, ranging from 14% among Q1 up to 20% for Q5. Among all drinker groups, mean consumption is either similar or demonstrates no clear pattern between wealth quintiles. On aggregate, there was a gradient in average baseline weekly spend with the rich paying an average R257.36 per week compared with R148.03 in the lowest wealth group.

Our model estimated, for an MUP of R10, an immediate reduction in population alcohol consumption of 4.40% (−0.93 SD/week) and an increase in spend of 18.09%. Moderate drinkers showed the greatest percentage decrease in their drinking, followed by occasional binge then heavy drinkers (−8.71%, −4.51%, −4.19%). However, this translated to a larger absolute reduction in consumption for heavier drinkers (−1.48 SD/week) than either occasional binge or moderate drinkers (−0.41 and to −0.40).

Our model estimated that there would be an increase in individual spend on alcohol consumption of R32.77 billion in the year following the introduction of the policy. The government would see an increase in VAT as a result of the increased prices although a reduction in excise taxation due to the reduced volume of alcohol sold. Retail revenue would also increase (table 3).

### Estimated health impact

Across the five health conditions included in the model, an R10 minimum price estimated 20 585 lives saved and 9 00 332 cases averted of the disease/injury conditions over the 20-year time horizon. For R5 (R15), we estimated 95 (45 326) lives saved and 4126 (2 038 319) cases averted, respectively. The impact differed by drinker group and by wealth quintile (figure 3). The greatest health benefits accrued to the heaviest drinkers, with the dominant effect related to HIV infections, especially in the bottom three

quintiles. Among the heavy drinkers, 85% of the cases averted and 86% of the lives saved accrued to the bottom three quintiles. Occasional binge drinkers achieved most of their positive health impact via a reduction in interpersonal violence and road injury as both of these conditions are linked to binge drinking. There was a small increase in HIV incidence among occasional binge drinkers. The high prevalence of HIV is the source of an important competing risk and the avoidance of death related to acute conditions led to longer exposure to the risk of HIV infection. As expected, the cases saved of liver cirrhosis accrued to the heavy drinkers, as this condition relates to heavy drinking in the long term. Q2 realised the highest number of HIV cases averted due to having the highest proportion of cases at baseline.

Healthcare cost savings accrued over the 20 years and were greatest for intentional injury (table 4). The health cost savings are provided by quintile in online supplemental appendix 16.

### Results across policy levels

Comparing across the three policy levels demonstrates the relative impact between wealth quintiles remained largely consistent as the MUP level increased for moderate and occasional binge drinkers (figure 4). For heavy drinkers the wealth gradient becomes more pronounced at R15 particularly with regards to the change in consumption.

The sensitivity analysis that produced the most variable results were the alternative elasticity estimates. Two of the alternative scenarios (−0.8 applied to all drinkers and −0.86/−0.5 applied to Q1 and Q2 with −0.5 applied to Q3 − Q5) produced much greater consumption impacts (−14%, − 18%) coupled with much smaller increases in individual spend (5.4%, 0.1%). All other results are included in online supplemental appendix 15.

## DISCUSSION

Our analysis estimates that MUP may offer an effective approach to reducing alcohol consumption and related harm in SA. For an MUP of R10, we estimate an immediate reduction in consumption of 4.40%, increase in individual spend of 18.09% and an increase in retail revenue and taxation. In terms of health impact, we estimate 20 585 lives saved and 9 00 332 cases averted in total across HIV, intentional injury, road injury, liver cirrhosis and breast cancer over 20 years. Regarding the equity impact, our model estimates that the distribution of health outcomes is generally pro-poor, critically important, given these groups also see the greatest relative increase in their alcohol expenditure.

Our research aligns with studies from other countries, which suggest that minimum pricing will reduce alcohol sales and also corresponds to mechanisms, such as greater impact with a rising MUP threshold and greater impact on the poor, found in the international literature.[24 25] We add to the South African minimum pricing evidence currently available[26] by incorporating health outcomes,

**Table 2** Consumption and spend R10 policy estimates

| | Overall | Q1 | Q2 | Q3 | Q4 | Q5 |
|---|---|---|---|---|---|---|
| Survey respondents | 10336 (100%) | 2098 (19%) | 2227 (19%) | 2337 (21%) | 2066 (20%) | 1608 (21%) |
| **All drinkers** | | | | | | |
| n (%)* | 3311 (33%) | 551 (27%) | 690 (30%) | 823 (33%) | 685 (35%) | 562 (38%) |
| Baseline consumption (standard drinks per week) | 21.22 | 20.83 | 21.40 | 20.97 | 21.98 | 20.89 |
| Baseline spending (R per week) | R208.74 | R148.03 | R192.82 | R186.95 | R231.78 | R257.36 |
| Change in consumption (%) | −4.40% | −7.75% | −6.42% | −3.76% | −3.41% | −3.19% |
| Change in consumption (standard drinks per week) | −0.93 | −1.50 | −1.29 | −0.76 | −0.72 | −0.65 |
| Change in spending (R per week) | R37.95 | R32.81 | R32.52 | R38.27 | R42.64 | R43.07 |
| **Moderate** | | | | | | |
| n (%)* | 1336 (12%) | 206 (10%) | 272 (13%) | 354 (12%) | 273 (13%) | 231 (15%) |
| Baseline consumption (standard drinks per week) | 5.05 | 5.01 | 5.49 | 4.90 | 4.86 | 4.98 |
| Baseline spending (R per week) | R49.97 | R42.75 | R48.84 | R43.54 | R54.38 | R56.59 |
| Change in consumption (%) | −8.71% | −12.20% | −12.89% | −7.14% | −6.35% | −6.43% |
| Change in consumption (standard drinks per week) | −0.40 | −0.55 | −0.63 | −0.33 | −0.29 | −0.30 |
| Change in spending (R per week) | R5.79 | R3.52 | R3.91 | R6.16 | R6.97 | R7.30 |
| **Occasional binge** | | | | | | |
| n (%)* | 433 (4%) | 76 (4%) | 89 (4%) | 109 (5%) | 91 (4%) | 68 (4%) |
| Baseline consumption (standard drinks per week) | 9.53 | 9.69 | 9.27 | 9.59 | 9.27 | 9.82 |
| Baseline spending (R per week) | R96.87 | R68.13 | R84.04 | R94.63 | R120.59 | R109.00 |
| Change in consumption (%) | −4.51% | −10.16% | −4.21% | −4.05% | −1.89% | −3.32% |
| Change in consumption (standard drinks per week) | −0.41 | −0.89 | −0.37 | −0.37 | −0.17 | −0.32 |
| Change in spending (R per week) | R14.58 | R16.69 | R9.28 | R17.86 | R11.31 | R16.42 |
| **Heavy** | | | | | | |
| n (%)* | 1542 (16%) | 269 (14%) | 329 (14%) | 360 (16%) | 321 (17%) | 263 (20%) |
| Baseline consumption (standard drinks per week) | 36.72 | 35.02 | 39.53 | 36.20 | 38.22 | 35.16 |
| Baseline spending (R per week) | R360.19 | R244.13 | R356.68 | R320.18 | R394.90 | R439.14 |
| Change in consumption (%) | −4.19% | −7.15% | −5.75% | −3.41% | −3.23% | −2.85% |
| Change in consumption (standard drinks per week) | −1.48 | −2.34 | −2.15 | −1.19 | −1.19 | −0.97 |
| Change in spending (R per week) | R69.68 | R57.17 | R65.17 | R67.85 | R77.47 | R75.08 |

Data for 10336 survey respondents.
*Numbers refer to absolute sample size, percentages incorporate survey weights, the relevant base is indicated in the top row of their column.
Q1, poorest.

accommodating homebrew and exploring differential impacts by wealth groups. Van Walbeek and Chelwa[26] who produced an economic model to simulate the impact of a MUP on consumption (with no epidemiological modelling) suggest both a higher reduction in consumption and a greater difference in consumption impact between heavy and moderate drinkers. The difference in our estimates is largely due to different price estimates. Their prices are crucially far more heterogeneous between drinker groups, outweighing the impact of the price elasticities. Our prices are drawn from a detailed survey asking for real prices paid by beverage, container and location, which allows us to calculate real prices per SD. Van Walbeek and Chelwa used an average unit value derived from reported monthly alcohol consumption (calculated using quantity/frequency questions) and one

**Table 3** Aggregate spend, taxation and retail revenue

Change from baseline in billion rand, per year

|  | R5 MUP | R10 MUP | R15 MUP |
|---|---|---|---|
| Individual spend | R1.24 | R32.77 | R78.29 |
| Taxation | | | |
| VAT | R0.16 | R4.27 | R10.21 |
| Excise tax | –R0.03 | –R1.24 | –R3.40 |
| Retail revenue | R1.11 | R29.74 | R71.48 |

MUP, minimum unit pricing; VAT, value-added tax.

**Table 4** Healthcare cost savings over 20 years, millions

|  | R5 MUP | R10 MUP | R15 MUP |
|---|---|---|---|
| Antiretroviral therapy costs | –R0.15 | R565.82 | R1356.51 |
| Intentional injury hospital costs | R32.55 | R4304.13 | R9088.97 |
| Road injury hospital costs | R16.46 | R1975.45 | R4265.68 |
| Liver cirrhosis hospital costs | R0.66 | R27.60 | R68.19 |
| Breast cancer hospital costs | R0.22 | R4.00 | R10.59 |

variable asking for monthly spend on alcohol,[26] which gave very low prices for heavy drinkers. Their prices may be too low and ours too high for the heaviest drinkers. If this is the case, our findings may present a conservative estimate of the potential impact of the policy.

Our study has a number of strengths relevant to providing policy-relevant research in LMICs. In the absence of detailed market research purchasing data, we demonstrate how survey, administrative data and the academic literature can be used, in partnership with local stakeholders, to build a contextually relevant epidemiological policy appraisal model. A further strength is our focus on stakeholder engagement from project inception increasing the likelihood of findings being taken into consideration during policy decision-making.[27] MUP was chosen as the policy to model as it was seen as both innovative and potentially well targeted for the South African heavy drinking culture. Stakeholders were pleased the estimates combined improved health with increased taxation and increased retail revenue, as supporting business was considered politically important. The financial cost of MUP is borne by drinkers and there were concerns about how this may impact poorer groups and we recommend this as an area for further research.

A limitation of our study is the lack of high-quality pricing data for SA. Previous studies in HIC have found that moderate drinkers, even those on lower incomes,

purchase relatively little cheap alcohol,[24] while the price data used in our model suggest that all drinker groups purchase some cheap alcohol. It is unclear whether this is a true reflection of alcohol purchasing patterns in SA or a limitation of the data. In addition, although we adjusted the off-trade wine prices to be consistent with industry sources, we know that the proportion of wine in the survey is less than the market share. As wine constitutes some of the cheapest available alcohol, an MUP may have a bigger impact than our estimates suggest. If the price of wine increased, we may expect drinkers to switch to other

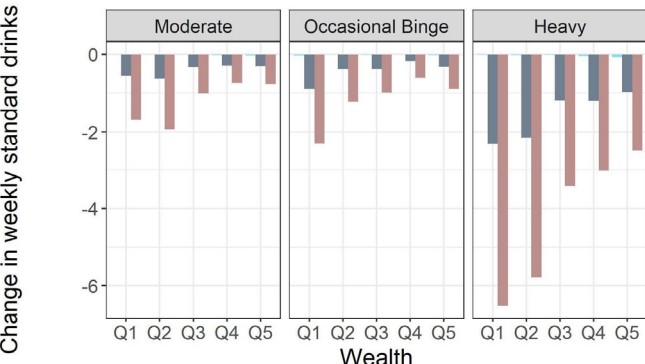

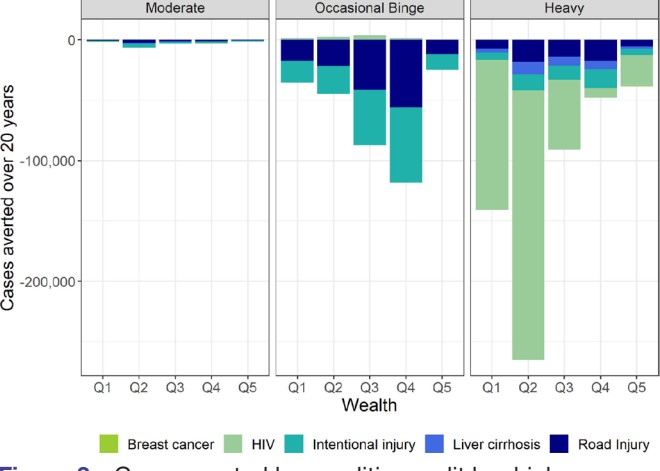

**Figure 3** Cases averted by condition, split by drinker group and wealth quintile.

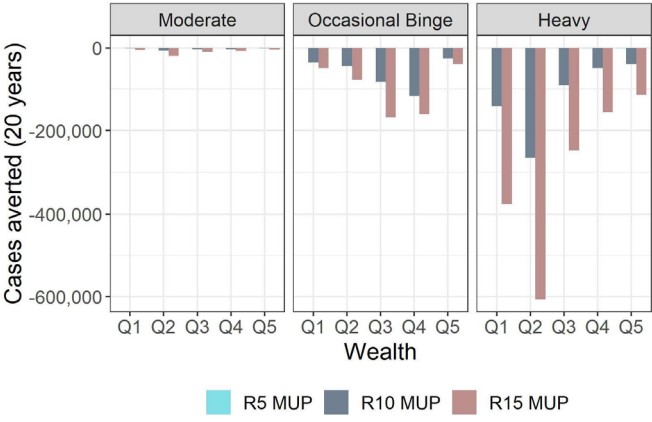

**Figure 4** Comparing the three policy levels: change in mean weekly drinks and cases averted by drinker and wealth group. MUP, minimum unit pricing.

cheaper alcohol, however, a key strength of MUP is that the policy applies across all alcohol types, and so drinkers are not able to do this.

Related to this, limitation is the treatment of all drinks as one commodity. South African evidence has suggested that cheap wine has a much higher price elasticity than other drink types.[8] If cheap wine is the drink primarily affected, then this elasticity would lead to less of an increase in individual spend, potentially even a saving, smaller gains to retailers and less of a loss to excise tax revenue as wine enjoys substantially lower rates.

We recommend the following avenues for further research. First, the collection of improved pricing data, specifically the different prices paid for alcohol by different population groups, to explore further the most appropriate level of MUP. Second, the exploration of the financial impact on the poorest groups including any financial benefits such as reduced expenditure on health-care or improved labour market outcomes. Third, in an alcohol market that includes retailers operating outside of the regulated space (despite largely selling recorded alcohol purchased from licensed outlets), it would be important to understand enforcement mechanisms and the supply chain in order for the policy to maximise effectiveness. However, it should be noted that the IAC pricing data suggest most of the lowest prices are to be found at large supermarkets and bottle stores, which offer bulk discounts rather than small local shebeens that sell alcohol often to be drunk on the premises.

## CONCLUSION

Our model estimates that minimum pricing would reduce alcohol consumption in SA, improving health outcomes while raising retail and tax revenue. Consumption and harm reductions would be greater in poorer compared with richer groups. We estimate that minimum pricing is a targeted policy that has the potential to bring health and financial benefits to a country, which suffers a very high burden of alcohol-related harm.

**Acknowledgements** We would like to thank all stakeholders who have shaped the research. This includes contributors from National Government, Western Cape Department of the Premier, Western Cape Liquor Authority, South African Alcohol Policy Alliance, Douglas Murray Trust, Khayelitsha Community Health Forum, Violence Protection through Urban Upgrading, Research Unit of the Economics of Excisable Products at University of Cape Town, South African Medical Research Council and the University of the Western Cape. We would also like to thank Charlotte Probst and Nick Stacey for their scientific review of the work.

**Contributors** All authors conceptualised the study. NG completed the modelling and stakeholder engagement under the supervision of CA, SD, PM and CP. NG wrote the first draft. All authors refined various drafts of the manuscript and approved the final version.

**Funding** This work was supported by the Wellcome Trust Doctoral Training Centre in Public Health Economics and Decision Science [108903/Z/19/Z] and the University of Sheffield [no award number]. Also the South African Medical Research Council [no award number]. The funders of the study had no role in the study. All authors had full access to all the data in the study and were responsible for the decision to submit the article for publication.

**Competing interests** None declared.

**Patient consent for publication** Not required.

**Ethics approval** Ethical approval for engaging with stakeholders was granted by the South African Medical Research Council (Protocol ID: EC005-4/2019) and the School of Health and Related Research at the University of Sheffield, UK (Reference Number: 023357). All data for the model came from secondary sources and were managed according to an approved information governance plan.

**Provenance and peer review** Not commissioned; externally peer reviewed.

**Data availability statement** Data may be obtained from a third party and are not publicly available. No additional data available.

**ORCID iDs**
Naomi Gibbs http://orcid.org/0000-0002-4704-8082
Colin Angus http://orcid.org/0000-0003-5529-4135

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
