## [Reviewer comments · BMJ Open]

ARTICLE DETAILS

TITLE (PROVISIONAL)	Effects of minimum unit pricing for alcohol in South Africa across different drinker groups and wealth quintiles: a modelling study.
AUTHORS	Gibbs, Naomi; Angus, Colin; Dixon, Simon; Parry, Charles; Meier, Petra

VERSION 1 – REVIEW

REVIEWER	Anita Lal Institute for Health Transformation, Deakin Health Economics
REVIEW RETURNED	13-Jun-2021

GENERAL COMMENTS	Comments to the authors General comments The study examines the health and expenditure effects of an increase in alcohol tax in South Africa by wealth quintile, a worthy area of research. There are several limitations of the analysis that reduce the accuracy of the findings and these are mentioned by the authors. Treating alcohol as one commodity here is clearly not ideal. The homebrew analysis attempts to capture a substitution effect but appears to be applied across the whole population. Could this be done more accurately by applying to the specific groups concerned? In the introduction it is mentioned that there are concerns regarding possible financial impact of pricing policies on the poorest groups however this is not further discussed in relation to the results found and needs to be addressed. How does the homebrew analysis impact this group especially? The authors mention the significance of shebeens and that these are largely unlicensed. If this is the case, how would MUPs be enforced? Further explanation is required. Abstract Objective: Suggest replace 'drinking' with 'alcohol consumption' Remove 'adult' as 15–17-year old's are not considered adults. Replace with: 'The population aged 15+' Line 32 remove 'then' Background Line 10: Should be leaders' Line 22 replace comma with a full stop. Paragraph 4 If the majority of the bars are unlicensed how will MUP have an impact there? Methods General comments The wealth quintile analysis is an important part of the modelling and should be included in the main methods section. There is no mention of this in the methods at present. What is your justification for not using education level groups rather than calculating wealth quintiles?
---

	It is ok to describe the model in the present tense but the majority of this section should be in the past tense as you are describing what you did. P6 line 50: Can you explain here a % change in price to make this clearer to the international readers what the relative size of the increase is? P7 line 10 What is the reference for the official sales volume data and why was this not used to calculate total retail spend? Line 47 What is IAC data? Expand initials and explain the survey please. What are the data constraints you refer to? This is a major assumption that needs to be made explicit. Price Elasticity of demand for alcohol also changes as income level changes. Using the same PE would underestimate the impact of the price increase. Explain how you have accounted for this using different price levels. Results General comment: This section should also be in the past tense where appropriate. P11 line 21 This sentence needs to be changed so that the model did the estimating not the R10 minimum price. It should also be in the past tense. Table 3 This table should be broken down into wealth quintiles to further emphasize the equity benefits of the policy. Using negative signs here is confusing. The only negative should be the first number, 0.15, as this is not a cost saving. Figure 3 The wealth quintile should be numbered 1 to 5 rather than the descriptions given (throughout the paper) P12 line 8 scenarios is misspelt. Explain what the alternative scenarios were. P45 graph Why is the HIV for Q2 so much higher than the rest? This needs an explanation. Conclusions Paragraph 1: The last sentence belongs in a separate paragraph around comparing your findings with other studies (the next paragraph) Paragraph 2 What did the Authors you mention analyse? Was it a similar policy? More details needed here. How did they determine your price estimates and why were yours so different from theirs? Can you make comparisons with other studies from other countries that have analysed similar policies eg Robinson et al 2020? Paragraph 3 and 4 should be combined. The opening sentence of Paragraph 3 leads the reader to believe there is more than one strength in this paragraph. P12 line 58 Presumably cheap wine drinkers switch to other drinks. What is the substitution effect happening here? Supplementary Tables P18 line 44 use of the term “ballpark” is too informal for a scientific paper. Avoid the use of “I” change to “we” to include your co-authors. 5. Graph needs an explanation on X axis. 13. Sensitivity analysis: Homebrew switching: why does 30% have more less lives saved than Scenario 2 100% switching?
--	---

REVIEWER	Akshar Saxena Nanyang Technological University
-----------------	---

GENERAL COMMENTS

Dear Editors and Authors,

Thank you for the opportunity to review this article.

The article conducts a simulation to ascertain the effect of changes in MUP for alcohol. It uses multi-state life-tables to evaluate the effect over 20 years.

The article is very well written and describes the methodology and results in great detail. I have some minor comments and suggestions appended below.

Comments for Authors:

Page 5 para 3 Background: Translation of policy evidence from the high-income countries to South-African setting: "A limitation of transferring..."?

- Authors may like to clarify further whether their intention is to highlight that translating evidence from high-income to low-income countries may be difficult due to the difference in proportion of household income spent on alcohol across the two sets of countries?

- Or is it only due to differences in epidemiology, and level of development across the two kinds of countries.

- Secondly, it is not clear whether authors imply that low-income countries, in general consume, higher levels of alcohol than high-income countries; or higher binge-drinking (Cutler et al indicate a high correlation between education and binge-drinking pattern)

- OR whether the trend across income-quintiles within a low-income country is likely to be different than the trend across income/wealth quintiles in a high-income country?

Page 6 para 5 "Applying a minimum unit price":

- May I suggest that the authors include Appendix 6 Table 8 somewhere in the main section. This is an important table as it provides context to the price-changes. The reader is able to ascertain which quintiles and groups of drinkers is likely to be affected.

- Was Table 8 or Table 9 finally used for simulation?

- In the South-African context, is the MUP pre-tax or post-tax?

Page 7 para 3 "This is adjusted by...". May I request the authors to clarify (in appendix) how did they do this adjustment ?

Page 9 para 2 "We estimate...". May I ask why do the authors switch between annual effect sizes in this paragraph but to weekly effect sizes in Table 1.

Page 10 Table 1:

- Table (Panel: All Drinkers) seems to indicate that the weekly spending for all wealth quintiles increases due to inelastic demand. Given the change in spending is roughly similar across the quintiles, then can we interpret that the change in weekly spending as proportion of income is the highest for the lowest wealth quintile while lowest for the highest quintile?

- If we only look at changes in alcohol expenditure, then the MUP seems exacerbate the inequality. It remains to be seen whether the difference in consumption (5.72% reduction vs 4.16% reduction) and resultant change in health and health-expenditure is able to compensate for this.

- The 1.64 percentage point difference (5.72% vs 4.16%) in consumption translates to roughly how many lives saved?

- Could the authors please provide numbers and comment on the reduction in mortality/morbidity and health-expenditure across quintiles and resultant saving/increase in total household expenditure by the wealth-quintiles?

- n(%). Could the authors please clarify as to how their %

	were calculated? The notes below the table seem inadequate. E.g. 10% indicated for Moderate Drinkers in Q1. This is 10% of what base? o It might be useful to have percentages of drinkers across quintiles. E.g. Within Q1, 30% are moderate, 30% are occasional binge while 40% are heavy. This will help the reader in obtaining a weighted average of the change in consumption, expenditure and health effects by quintiles. Minor typographical issues: Page 9 para 1 Results: “Our findings are presented primarily for R10”. May I suggest that the authors include “MUP of” before R10 as at first glance it may not be obvious to the reader that it is referring to policy scenario of RAND10. Also, briefly indicate how much %age increase in price would this be. Page 11 para 1 “Our model estimates that there would be an increase in individual spend of R32.06 billion in the year following the introduction of the policy.” May I suggest that the authors include the phrase “on alcohol consumption” in the sentence to ensure that readers do not misinterpret this figure as the final figure after netting out the effect of health-savings.
--	---

VERSION 1 – AUTHOR RESPONSE

Reviewer One

General comments

The study examines the health and expenditure effects of an increase in alcohol tax in South Africa by wealth quintile, a worthy area of research. There are several limitations of the analysis that reduce the accuracy of the findings and these are mentioned by the authors. Treating alcohol as one commodity here is clearly not ideal. The homebrew analysis attempts to capture a substitution effect but appears to be applied across the whole population. Could this be done more accurately by applying to the specific groups concerned?

We would like to sincerely thank reviewer one for their considered and helpful notes on our paper. They were specific and constructive and we hope our response to them has improved both the quality and transparency of our work. We thank them for their kind words about the worthiness of our research. We agree, as noted in the manuscript, on the limitations of the data which we hope might strengthen the argument that detailed and accurate pricing data would greatly contribute to the generation of empirical evidence to inform alcohol policy in South Africa.

We note the reviewer’s concern about the homebrew analysis, however they are incorrect in their understanding that this is applied at the whole population level. In fact the amount of homebrew that is drunk is estimated at an individual level and comes from self-reports in the South African Demographic and Health Survey Data. We have now amended the text to make this process clearer.

Page 5 lines 14 - 18

I. “Price to consumption: Baseline prices are estimated for drinker groups (heavy drinkers, occasional binge drinkers, moderate drinkers) and wealth groups. Consumption is estimated at the individual level, this includes the proportion of alcohol drunk that is homebrew. Following a change in price the new price and subsequent consumption levels are estimated. This accounts for both mean and peak weekly alcohol consumption. “

In the introduction it is mentioned that there are concerns regarding possible financial impact of pricing policies on the poorest groups however this is not further discussed in relation to the results found and needs to be addressed. How does the homebrew analysis impact this group especially?

We appreciate the reviewer's point which highlights the importance of the policy on the poorest groups. We agree that the financial impacts on the poorest groups are very interesting and important, but our stakeholders were keenly interested in the consumption and health aspects of the results as well as the tax and retail revenue, which is why we have chosen this focus for this paper. We are therefore undertaking an Extended Cost Effectiveness Analysis, using the same model, which explores financial risk protection and labour costs, in a separate publication.

We feel this paper provides the right level of detail, demonstrating the policy and the model and keeping it broad for a wide policy audience. As the homebrew represents a small proportion of the consumption it has a limited impact on the model results as we have demonstrated in the sensitivity analysis in the appendix part 15.

The authors mention the significance of shebeens and that these are largely unlicensed. If this is the case, how would MUPs be enforced? Further explanation is required.

We thank the reviewer for this helpful point. It is a highly complex issue and one that we know policy makers in the Western Cape Province of South Africa are currently grappling with. Although we do not have the relevant expertise in this area, or scope to do the topic justice, we have included the following two sentences in the final paragraph to provide further explanation to provide greater context for the reader.

Page 12 lines 18 - 23

Thirdly, in an alcohol market that includes retailers operating outside of the regulated space (despite largely selling recorded alcohol purchased from licensed outlets) it would be important to understand enforcement mechanisms and the supply chain in order for the policy to maximise effectiveness. However, it should be noted that the IAC pricing data suggests most of the lowest prices are to be found at supermarkets and bottle stores which offer bulk discounts rather than small local shebeens that sell alcohol often to be drunk on the premises.

Abstract

Objective: Suggest replace 'drinking' with 'alcohol consumption'

We have amended the text on page 2 line 3

Remove 'adult' as 15–17-year old's are not considered adults. Replace with: 'The population aged 15+'

We have amended the text on page 2 lines 11 and 13.

Line 22 remove 'then'

We have amended the text on page 2 line 22

Background

Line 10: Should be leaders'

We have amended the text on page 4 line 6

Line 32 replace comma with a full stop.

We have amended the text

Paragraph 4 If the majority of the bars are unlicensed how will MUP have an impact there?

We hope we have addressed this now in our response to your point above. Please refer to Page 12 lines 18 - 23 for the additional sentences.

Methods

General comments

The wealth quintile analysis is an important part of the modelling and should be included in the main methods section. There is no mention of this in the methods at present. What is your justification for not using education level groups rather than calculating wealth quintiles?

We appreciate the reviewer highlighting how important the wealth quintile analysis is to our work. We used wealth quintiles as we needed to match the consumption estimates (arising from the SADHS dataset) with measures which were available to us in the IAC dataset (income was not) in order to be able to demonstrate that the baseline price paid varied by wealth (or some measure of SES). This was important as baseline health is also linked to SES and we wanted to be able to demonstrate differential policy impact. Education is one of the variables we have included in the ordered choice regression model. We have outlined the rationale and methods used in Appendix 4 which now reads:

Appendix part 4

In order to match wealth groups between the two datasets an ordered choice model was created using SADHS data with wealth quintile (1 – 5) as the dependent variable, using the MASS package in R. Wealth groups were chosen as the best available measure to capture socioeconomic status that allowed us to match between the SADHS and IAC dataset. Although income was asked in the IAC dataset many of the respondents refused to answer resulting in a very small sample.

All the variables that were common across the two datasets were included in the initial model, these were not just asset ownership but also age, sex, educational level and population group (race).

Stepwise regression was performed using the step.AIC function. This chooses the best variables to include by running the regression with all variables in and then taking one out and computing a goodness of fit measure (the AIC). If the goodness of fit measure is improved then that model is preferred, it runs this for many models until it finds the model with the highest AIC. This method resulted in the selection of the following variables: age, sex, population group, education level, car, landline, electricity, fridge, computer, radio, tv. The only variable it removed was mobile phone which fitted anecdotally with conversations I had with stakeholders in South Africa regarding how much poorer people prioritise mobile phones.

We have also added the following sentences to the main methods section in the text

Page 5 lines 40 - 42

Wealth quintiles were chosen as our measure of socioeconomic status as income was not available in the pricing dataset whereas asset ownership and common demographic data such as age, sex and education was.

It is ok to describe the model in the present tense but the majority of this section should be in the past tense as you are describing what you did.

Thank you for highlighting this and we agree. We have now rewritten the methods section to be in the past tense.

P6 line 50: Can you explain here a % change in price to make this clearer to the international readers what the relative size of the increase is?

We were initially reticent to provide a percentage increase in mean price as we did not want to mislead people in thinking that a MUP creates a percentage increase in price across all prices. We hope we have explained the policy mechanism sufficiently and have now inserted a table with mean base price, percentage change in mean price and the elasticities used in the methods section.

Page 6 lines 1 - 15

Applying a minimum unit price

A government policy of legislating for a MUP of R5, R10 and R15 per South African standard drink was introduced. Prices below the MUP threshold were increased to the threshold, while products above were unaffected. We did not include prices for homebrew. The distribution of prices faced by each wealth/drinker group was used to calculate the mean price per standard drink before and after the policy. This then provided a percentage change in mean price (Table 1).

Elasticity of demand for alcohol

The change in price was translated into a change in individual consumption using an elasticity of demand for alcohol. We used previously published elasticities for South Africa, calculated separately by drinker group: -0.4, -0.22, -0.18 for moderate, occasional binge and heavy drinkers respectively and adjusted for wealth quintile using additional evidence from South Africa (Table 1) (Appendix).

Table 1: Price per standard drink (15ml of pure ethanol) before the policy by wealth quintile and drinker group

	Q1	Q2	Q3	Q4	Q5	
Baseline prices						
Moderate		R9.13	R9.13	R9.13	R11.6	R11.6
Occasional Binge			R7.97	R10.0	R10.1	R13.4 R11.1
Heavy	R7.78	R9.65	R9.23	R10.6	R12.8	
Percentage change in mean price following R10 MUP						
Moderate		22%	22%	22%	20%	20%
Occasional Binge			37%	16%	24%	11% 19%
Heavy	33%	26%	25%	24%	21%	
Price elasticities used in the model						
Moderate		-0.53	-0.53	-0.31	-0.31	-0.31
Occasional Binge			-0.29	-0.29	-0.17	-0.17 -0.17
Heavy	-0.24	-0.24	-0.14	-0.14	-0.14	

Q1 = poorest

P7 line 10 What is the reference for the official sales volume data and why was this not used to calculate total retail spend?

We would like to thank the reviewer for this comment. We have now added the reference for the official sales volume data which comes from Euromonitor International Passport. We calculated retail spend using the IAC pricing data and quantities which were calibrated to total consumption. If we wanted to calibrate to total sales revenue we would have to make too many unsubstantiated assumptions about the shape of the pricing distributions.

Line 47 What is IAC data? Expand initials and explain the survey please.

We have expanded the initials, added some more details and provided a reference to the survey webpage.

Page 5 lines 34 - 40

We generated price distributions for wealth and drinker groups using real price data linked to individual drinking from the International Alcohol Control Study (IAC) survey 2014/2015 completed in the metropolitan district of Tshwane. The IAC asked for highly detailed data about prices in both on-

and off-trade locations and took into account container size, drink type and number of drinks purchased.

What are the data constraints you refer to? This is a major assumption that needs to be made explicit. Price Elasticity of demand for alcohol also changes as income level changes. Using the same PE would underestimate the impact of the price increase. Explain how you have accounted for this using different price levels.

Thank you for highlighting the ambiguity of the sentence highlighting data constraints. We have now replaced the sentence with the following:

Page 5 lines 38 - 39

Alcohol was treated as one commodity as the 863 price observations were distributed between drinker and wealth groups instead of by alcohol type.

With regard to the reviewer's questions relating to our use of price elasticities which do not change with income level (or in our study wealth level), we have consulted both the research available to us and the views of local stakeholders with academic expertise in economics of excisable products and have changed our elasticities as follows. We now have a 3 x 5 matrix of elasticities which accommodates both drinker and wealth group. We have provided an explanation for the adjustment in the appendix and have explained it in the main body of the text. All the results, tables and figures were revised to reflect this. We include the elasticities which we had been using previously as our base case now as a sensitivity analysis.

Page 6 lines 8 - 13

Elasticity of demand for alcohol

The change in price was translated into a change in individual consumption using an elasticity of demand for alcohol. We used previously published elasticities for South Africa, calculated separately by drinker group: -0.4, -0.22, -0.18 for moderate, occasional binge and heavy drinkers respectively and adjusted for wealth quintile using additional evidence from South Africa (Table 1) (Appendix 7).

	Q1	Q2	Q3	Q4	Q5	
Baseline prices						
Moderate		R9.13	R9.13	R9.13	R11.6	R11.6
Occasional Binge			R7.97	R10.0	R10.1	R13.4 R11.1
Heavy	R7.78	R9.65	R9.23	R10.6	R12.8	
Percentage change in mean price following R10 MUP						
Moderate		22%	22%	22%	20%	20%
Occasional Binge			37%	16%	24%	11% 19%
Heavy	33%	26%	25%	24%	21%	
Price elasticities used in the model						
Moderate		-0.53	-0.53	-0.31	-0.31	-0.31
Occasional Binge			-0.29	-0.29	-0.17	-0.17 -0.17
Heavy	-0.24	-0.24	-0.14	-0.14	-0.14	

Q1 = poorest

Appendix part 7

The starting point for elasticities -0.4, -0.22 and -0.18 for moderate, occasional binge and heavy drinkers respectively. We adjusted these elasticities to incorporate an income gradient using -0.86 and -0.5 elasticity for low and high socioeconomic status. To remain on the conservative side we will count the bottom two quintiles as low SES and the top three as high.

Table 10: Elasticities by wealth and drinker group

Drinker type	Q1	Q2	Q3	Q4	Q5
Moderate	-0.53	-0.53	-0.31	-0.31	-0.31
Occasional binge	-0.29	-0.29	-0.17	-0.17	-0.17
Heavy drinkers	-0.24	-0.24	-0.14	-0.14	-0.14

Results

General comment: This section should also be in the past tense where appropriate.

We have now rewritten the results section to be in the past tense.

P11 line 12 This sentence needs to be changed so that the model did the estimating not the R10 minimum price. It should also be in the past tense.

We have modified the text to read:

Page 8 lines 10 - 11

“Our model estimated, for a MUP of R10, an immediate reduction in population alcohol consumption of 4.50% (-0.95 SD/week) and an increase in spend of 17.83%.”

Table 3 This table should be broken down into wealth quintiles to further emphasize the equity benefits of the policy. Using negative signs here is confusing. The only negative should be the first number, 0.15, as this is not a cost saving.

We agree with your point that the negative signs are confusing so have changed this. We feel that keeping the figures at an aggregate level in this table is more consistent with the table on tax and retail revenue which precedes it. We also know from engagement that policy makers are interested in the high level figures, however, we agree that it is useful to understand the equity benefits further and this have included the table you suggested in the appendix for those readers who are interested.

Appendix 16

Table 16: Health care costs for each of the three policy scenarios split by wealth quintile

	Q1	Q2	Q3	Q4	Q5
R5 MUP					
HIV	-R0.01	-R0.07	-R0.04	-R0.03	-R0.01
Intentional injury			R1.41	R5.22	R5.42
Road injury	R0.71	R2.73	R2.80	R6.39	R3.82
Liver cirrhosis	R0.02	R0.12	R0.11	R0.27	R0.14
cancer	R0.00	R0.00	R0.01	R0.05	R0.15
R10 MUP					
HIV	R162.00		R291.00	R71.10	R8.72
Intentional injury			R495.57	R801.23	R1150.94
Road injury	R232.98		R399.34	R520.70	R658.80
Liver cirrhosis	R3.03	R9.64	R6.62	R6.45	R1.86
cancer	R0.30	R0.22	R0.80	R0.93	R1.75
R15 MUP					
HIV	R403.19		R618.29	R190.50	R79.85
Intentional injury			R1136.23	R2029.50	R2558.09
Road injury	R536.83		R1013.46	R1173.35	R1080.17
Liver cirrhosis	R7.42	R23.50	R17.60	R15.20	R4.51

cancer R0.76 R0.65 R2.24 R2.30 R4.65

Figure 3 The wealth quintile should be numbered 1 to 5 rather than the descriptions given (throughout the paper)

Wealth quintiles have been changed to Q1 to Q5 across all tables and figures and in the main body of the text.

P12 line 8 scenarios is misspelt. Explain what the alternative scenarios were.

We have corrected this misspelling and also now spell out in the text what the scenarios are.

Page 11 lines 3 - 7

The sensitivity analysis that produced the most variable results were the alternative elasticity estimates. Both alternative scenarios (-0.8 applied to all drinkers, and -0.86/-0.5 applied to quintile 1 and 2 with -0.5 applied to quintiles 3 – 5) produced much greater consumption impacts (-14%, – 18%) coupled with much smaller increases in individual spend. All other results are included in appendix 14.

P45 graph Why is the HIV for Q2 so much higher than the rest? This needs an explanation.

To address this concern we have added the following explanation to the text.

Page 10 lines 20 - 21

Q2 realised the highest number of HIV cases averted due to having the highest proportion of cases at baseline.

Conclusions

Paragraph 1: The last sentence belongs in a separate paragraph around comparing your findings with other studies (the next paragraph)

We agree that this last sentence does not belong at the end of the first paragraph and have moved it to the second paragraph where we compare our work with other studies.

Paragraph 2 What did the Authors you mention analyse? Was it a similar policy? More details needed here. How did they determine their price estimates and why were yours so different from theirs? Can you make comparisons with other studies from other countries that have analysed similar policies eg Robinson et al 2020?

We have added explanation around what the policy is, and the methodological and data differences in results. Please note we have also updated the reference as their work has now been published in the South African Medical Research Journal.

Page 11 lines 21 – 32

Van Walbeek and Chelwa (2021) who produced an economic model to simulate the impact of a MUP on consumption (with no epidemiological section to the model) suggest both a higher reduction in consumption and greater difference in consumption impact between heavy and moderate drinkers. The difference in our estimates is due to different price estimates. Their prices are crucially far more heterogeneous between drinker groups, outweighing the impact of the price elasticities. Our prices are drawn from a detailed survey asking for real prices paid by beverage, container and location which allows us to calculate real prices per standard drink. Van Walbeek and Chelwa 2021 used an average unit value derived from reported monthly alcohol consumption (calculated using quantity/frequency questions) and one variable asking for monthly spend on alcohol which gave very low prices. Their prices are likely too low and ours too high for the heaviest drinkers. If this is the case our findings may present a conservative estimate of the potential impact of the policy.

In quoting Robinson et al 2020 we presume you mean the following report:

<https://publichealthscotland.scot/media/3093/evaluating-the-impact-of-mup-on-sales-based-alcohol-consumption-in-scotland-controlled-interrupted-time-series-analyses.pdf>

Which was subsequently updated to this report:

<https://publichealthscotland.scot/media/3094/evaluating-the-impact-of-mup-on-sales-based-alcohol-consumption-in-scotland-controlled-interrupted-time-series-analyses-briefing-updated-march-2021.pdf>

And was finally published in Addiction here:

<https://onlinelibrary.wiley.com/doi/full/10.1111/add.15478>

We have amended the following sentence and added the suggested reference to our pre-existing reference.

Page 11 lines 17 - 19

Our research aligns with studies from other countries which suggest minimum pricing will reduce alcohol sales and also corresponds to mechanisms, such as greater impact with a rising MUP threshold and greater impact on the poor, found in the international literature.

Paragraph 3 and 4 should be combined. The opening sentence of Paragraph 3 leads the reader to believe there is more than one strength in this paragraph.

We agree and have now combined paragraph 3 and 4 in the manuscript.

P12 line 58 presumably cheap wine drinkers switch to other drinks. What is the substitution effect happening here?

Thank you for highlighting substitution effects which are often included in alcohol modelling but have not been here as we treat alcohol as one commodity. It is a strength of the policy that drinkers cannot substitute to other drinks at a lower price per standard drink as the MUP applies to all alcohol types. We have added the following sentence

Page 12 line 6 - 8

If the price of wine increased we may expect drinkers to switch to other cheaper alcohol, however, a key strength of MUP is that the policy applies across all alcohol types and so drinkers are not able to do this.

Supplementary Tables

P18 line 44 use of the term "ballpark" is too informal for a scientific paper.

We have corrected it so that it now reads:

Appendix part 1

"this is of a similar magnitude"

Avoid the use of "I" change to "we" to include your co-authors.

We have changed the use of "I" to "We" in every location I could identify it. There were five occurrences that we found.

5. Graph needs an explanation on X axis.

We have added explanation regarding what a standard drink consists of and defined on and off trade in the figure legend. It now reads as follows:

Appendix part 5

Figure 5: Distribution of off-trade and on-trade prices, standard drink is 15ml or 12grams of pure ethanol.

On-trade is where the alcohol is consumed on the premises it is purchased (e.g. hotels, restaurants, pubs), off-trade is where the alcohol is consumed off the premises it was purchased at (e.g. supermarket or bottle store).

13. Sensitivity analysis: Homebrew switching: why does 30% have more less lives saved than Scenario 2 100% switching?

We have modified the text and hope that we have made it clearer how the homebrew substitution works. In essence drinkers who drink both recorded alcohol and homebrew will reduce their recorded alcohol consumption when faced with a price rise but may simply drink a bit more homebrew. For example if someone drinks 10 fewer drinks of recorded alcohol as a result of the policy they may increase their homebrew drinks by 3 (30%) which dampens the impact of the policy. If they increased their homebrew by 100% of the reduction then although they reduce their recorded alcohol consumption by 10 drinks they increase their homebrew drinking by 10 drinks (100% switching). Therefore, there is no reduction in consumption as a result of the policy for this group and that is why there are less lives saved.

Page 6 lines 16 - 19

Those that drink both recorded alcohol and homebrew dampened the policy impact by switching some of their drinking to homebrew. Stakeholders indicated 30% of the reduction in recorded alcohol could be assumed as being compensated for via an equivalent increase in homebrew. This was varied between 0% (no switching) and 100% (full switching) in the sensitivity analysis.

Reviewer two

We would like to thank reviewer two for the time they took reading and providing very helpful notes on how to improve the paper.

Background Paragraph 3

Page 5 para 3 Background: Translation of policy evidence from the high-income countries to South-African setting: "A limitation of transferring...?"

- Authors may like to clarify further whether their intention is to highlight that translating evidence from high-income to low-income countries may be difficult due to the difference in proportion of household income spent on alcohol across the two sets of countries?
- Or is it only due to differences in epidemiology, and level of development across the two kinds of countries.
- Secondly, it is not clear whether authors imply that low-income countries, in general consume, higher levels of alcohol than high-income countries; or higher bingedrinking (Cutler et al indicate a high correlation between education and bingedrinking pattern)
- OR whether the trend across income-quintiles within a low-income country is likely to be different than the trend across income/wealth quintiles in a high-income country?

We have edited this paragraph in an attempt to clarify it.

In response to the first point we did not analyse the proportion of household income spent on alcohol, however the income inequality does result in different prices and also different price responsiveness. We do wish to highlight that there are also important epidemiological differences, how much people abstain/binge drink and the alcohol harm profile of the country.

We do not make any broader statements about low-income countries drinking more or less or binge drinking more or less than high-income countries as there will be a great deal of variation within LMICs, we instead focus on South Africa which has a very clear drinking pattern in terms of abstinence and binge drinking. Our analysis is not to compare in general high-income with low-income countries but to focus on what is unique in the context of South Africa which is likely to impact the policy appraisal although we do believe there are lessons for alcohol research in the absence of

transaction level purchasing data which may be a common challenge for LMIC research of this kind. We have edited the paragraph and hope we have now been able to make this clearer to the reader:
Page 4 lines 15 - 19

A limitation of transferring the current evidence for MUP is its focus on high income countries. Transferring this evidence to South Africa would be problematic as it has very different drinking patterns, a very different harm profile with infectious disease and injury contributing significantly to the burden of alcohol, it has an informal sector which is challenging to capture and it has very high levels of income inequality likely to result in differential baseline prices and price responsiveness.

Page 6 para 5 “Applying a minimum unit price”:

- May I suggest that the authors include Appendix 6 Table 8 somewhere in the main section. This is an important table as it provides context to the price-changes. The reader is able to ascertain which quintiles and groups of drinkers is likely to be affected.

- Was Table 8 or Table 9 finally used for simulation?

We have now moved Table 9 to the main manuscript. We agree with the reviewer that this provides the reader with valuable context. In response to our other reviewer we also provided further details in this table which we hope improve it. Table 9 was the final table used for simulation as it ensured an adequate sample size for the prices. We have also made this clearer in the appendix through adding a sentence to the text above the table.

Page 6 line 14 - 15

Table 1: Price per standard drink (15ml of pure ethanol) before the policy by wealth quintile and drinker group

	Q1	Q2	Q3	Q4	Q5	
Baseline prices						
Moderate		R9.13	R9.13	R9.13	R11.6	R11.6
Occasional Binge			R7.97	R10.0	R10.1	R13.4 R11.1
Heavy	R7.78	R9.65	R9.23	R10.6	R12.8	
Percentage change in mean price following R10 MUP						
Moderate		22%	22%	22%	20%	20%
Occasional Binge			37%	16%	24%	11% 19%
Heavy	33%	26%	25%	24%	21%	
Price elasticities used in the model						
Moderate		-0.53	-0.53	-0.31	-0.31	-0.31
Occasional Binge			-0.29	-0.29	-0.17	-0.17 -0.17
Heavy	-0.24	-0.24	-0.14	-0.14	-0.14	

Q1 = poorest

Supplementary Appendix part 6

This represents the final group of prices used in the model.

In the South-African context, is the MUP pre-tax or post-tax?

The MUP is a retail floor price therefore the retailers would only be able to calculate VAT once they have applied the price, excise tax would be calculated as before as this relates to the alcohol content or the volume of alcohol (depending on the drink type as South Africa has a varied tax system). In order to clarify this for the reader we have added the following sentence:

Page 6 line 23

As a MUP is applied before VAT is calculated we estimate VAT as 15% of the total aggregate spend.

Page 7 para 3 “This is adjusted by...”. May I request the authors to clarify (in appendix) how did they do this adjustment?

We have inserted an extra section into the appendix explaining the steps taken in the calculation and have referenced this in the main text.

Page 7 paragraph lines 23 - 27

As a MUP is applied before VAT is calculated we estimate VAT as 15% of the total aggregate spend. Excise tax is calculated by starting with the total 2018 alcohol excise tax revenue from the Treasury Budget Report. This is adjusted by percentage change in volume of alcohol sold (we use a fixed ratio between volume and excise tax). Retail revenue is calculated by taking VAT and excise taxation away from total spend (Appendix 8).

Page 9 para 2 "We estimate...". May I ask why do the authors switch between annual effect sizes in this paragraph but to weekly effect sizes in Table 1.

We have now changed the figures in the text to weekly figures in order to match Table 1. We have also updated the abstract to this effect.

Page 2 lines 20 - 23

We estimate a MUP of R10 per SD would lead to an immediate reduction in consumption of 4.40% (-0.95 SD/week) and an increase in spend of 18.09%. The absolute reduction is greatest for heavy drinkers (-1.48 SD/week), followed by occasional binge drinkers (-0.41 SD/week) and moderate drinkers (-0.40 SD/week). Over 20 years we estimate 20,585 fewer deaths and 900,332 cases averted across the five health modelled harms.

Page 10 Table 1: Table (Panel: All Drinkers) seems to indicate that the weekly spending for all wealth quintiles increases due to inelastic demand. Given the change in spending is roughly similar across the quintiles, then can we interpret that the change in weekly spending as proportion of income is the highest for the lowest wealth quintile while lowest for the highest quintile?

We thank reviewer two for their question regarding the regressivity or otherwise of this policy. Please see the new results given the changing elasticities. Table 2 demonstrates that the change in consumption now increases with wealth however, given South Africa's very high income elasticity we know that this increase will still be greater as a percentage of income for the poorest groups. We do not wish to focus on this aspect of the policy too much beyond flagging the importance of exploring the financial impact of the poor in further research. We feel it is a larger topic that requires an additional study (which we are working on) and we would like to maintain the focus of this paper in explaining in detail the methodology of the model and highlighting the differential impact by drinker group, as much as by wealth quintiles.

If we only look at changes in alcohol expenditure, then the MUP seems exacerbate the inequality. It remains to be seen whether the difference in consumption (5.72% reduction vs 4.16% reduction) and resultant change in health and health-expenditure is able to compensate for this.

We thank reviewer two for their analysis of the results. As a result of the changes suggested by reviewer one the gradient in change in consumption is now from 7.75% for Q1 and 3.19% for Q5. It is not our purpose within this paper to directly quantify whether the increased alcohol expenditure is compensated for by the health and health expenditure savings. We leave this for further analysis which we are currently undertaking to complete and ECEA using this model. Our focus is on providing broad level policy results which highlight the differential policy effect and crucially health benefits will accrue to heavier drinkers, particularly amongst the poor.

The 1.64 percentage point difference (5.72% vs 4.16%) in consumption translates to roughly how many lives saved?

As mentioned above these numbers have now been revised and there is a steeper gradient between Q1 and Q5. Your question would now apply to the difference between 7.75% and 3.19%. It is not possible to quantify how many lives saved a 1.64 (or a 4.56) percentage point difference makes as it is not linear. The baseline risk relates to an individual's wealth and drinker type, to be clear HIV is concentrated amongst the poorer groups, intentional injury, road injury and liver cirrhosis amongst the middle and breast cancer amongst the richest. All of these conditions have different shaped relative risk curves also, or in the case of HIV a stepped function. Potential impact fractions are calculated based on wealth and drinker group. As the model runs it is also the case that lives saved from intentional injury or road injury may go on to be lost to HIV or at least increase the prevalence. If we

had used on all-cause alcohol attributable risk measure we may have been able to answer your question but even then it would have been very non-linear so the relative starting points on the curve would need to be very clear.

Could the authors please provide numbers and comment on the reduction in mortality/morbidity and health-expenditure across quintiles and resultant saving/increase in total household expenditure by the wealth-quintiles?

We have chosen not to focus so much on wealth quintiles (although we are working on this analysis separately) in this paper as the focus of the stakeholders was very much on drinker groups, with a particular concern for reducing heavy and binge drinking. We would need to make a number of additional assumptions to provide the data you request (such as the split between government and individual payment). We have provided an additional table in the appendix which breaks down the healthcare savings by quintile for readers who are particularly interested.

n(%). Could the authors please clarify as to how their % were calculated? The notes below the table seem inadequate. E.g. 10% indicated for Moderate Drinkers in Q1. This is 10% of what base?

It might be useful to have percentages of drinkers across quintiles. E.g. Within Q1, 30% are moderate, 30% are occasional binge while 40% are heavy. This will help the reader in obtaining a weighted average of the change in consumption, expenditure and health effects by quintiles.

We have clarified the table by including an additional row at the top which gives the number within each quintile and added a footnote to the bottom of the table. We hope it is now clear that for example by looking down the Q1 column the reader can see that 14% are heavy drinkers, 4% occasional binge and 10% moderate.

Table 2 footnote reads:

*Numbers refer to absolute sample size, percentages incorporate survey weights, the relevant base is indicated in the top row of their column.

Minor typographical issues:

Page 9 para 1 Results: “Our findings are presented primarily for R10”. May I suggest that the authors include “MUP of” before R10 as at first glance it may not be obvious to the reader that it is referring to policy scenario of RAND10. Also, briefly indicate how much %age increase in price would this be.

We agree that it is important to make clear that we are modelling a MUP. We have changed the sentence of page 9 para 1 to read

“Our findings are presented primarily for a R10 MUP but with some comparisons across all three pricing levels”

The percentage change in price is dependent upon the underlying distributions which provide the mean prices as outlined in Table 9 of the appendix (now added into the main body of the text). We have also now provided the percentage change in price to the same table which is on Page 6 line 14 – 15, Table 1.

Page 11 para 1 “Our model estimates that there would be an increase in individual spend of R32.06 billion in the year following the introduction of the policy.” May I suggest that the authors include the phrase “on alcohol consumption” in the sentence to ensure that readers do not misinterpret this figure as the final figure after netting out the effect of health-savings.

We have changed the text to the following for Page 11 paragraph 1

“Our model estimates that there would be an increase in individual spend on alcohol consumption of R32.77 billion in the year following the introduction of the policy.”

VERSION 2 – REVIEW

REVIEWER	Anita Lal Institute for Health Transformation, Deakin Health Economics
REVIEW RETURNED	27-Jul-2021
GENERAL COMMENTS	Well done on thoroughly addressing all of the reviewers'

	suggestions.
REVIEWER	Akshar Saxena Nanyang Technological University
REVIEW RETURNED	27-Jul-2021
GENERAL COMMENTS	Thank you for the extensive revision and addressing the points raised by the reviewers. Looking forward to reading the subsequent study on regressivity and distribution issues.